# Effects of Infant Formula Type on Early Childhood Growth Outcomes: A Retrospective Cohort Study [note 1]

**DOI:** 10.3390/nu17193111

**Published:** 2025-09-30

**Authors:** Uzma Rani, Roba Alwasila, William T. Story, Patrick Ten Eyck, Asher Hoberg, Donna A. Santillan, Aamer Imdad

**Affiliations:** 1Stead Family Department of Pediatrics, Division of General Pediatrics, Carver College of Medicine, University of Iowa, Iowa City, IA 52242, USA; uzma-rani@uiowa.edu; 2Division of Gastroenterology, Hepatology, Pancreatology, and Nutrition, Stead Family Department of Pediatrics, Carver College of Medicine, University of Iowa, Iowa City, IA 52242, USA; roba-alwasila@uiowa.edu; 3Department of Community and Behavioral Health, College of Public Health, University of Iowa, Iowa City, IA 52242, USA; william-story@uiowa.edu; 4Institute for Clinical and Translational Science, University of Iowa, Iowa City, IA 52242, USA; patrick-teneyck@uiowa.edu (P.T.E.); asher-hoberg@uiowa.edu (A.H.); 5Department of Obstetrics and Gynecology, University of Iowa, Iowa City, IA 52242, USA; donna-santillan@uiowa.edu

**Keywords:** infant nutrition, breastfeeding, growth

## Abstract

**Objective**: This study examines the effects of non-standard (lactose-reduced, hydrolyzed), cow-milk-based infant formulas on early childhood growth outcomes compared to standard formulas and breastfeeding. **Methods**: This retrospective cohort study included full-term infants with a birth weight >2500 g. Exposure and control data, including the type of infant formula [non-standard vs. standard] and breastfeeding status, were obtained at 2-month well visits. Growth outcomes (weight-for-age, length-for-age, BMI (Body Mass Index), and weight-for-length z-scores] were calculated using WHO (World Health Organization) growth standards at 1- and 2-year well visits. Generalized linear mixed models were used to evaluate associations between formula type and growth outcomes, adjusting for maternal, infant, and socioeconomic factors. **Results**: A total of 5515 infants were included in the final analysis. Feeding practices included exclusive breastfeeding (35%), standard formula (42%), and non-standard formula (23%). Infants fed non-standard formulas had significantly higher weight-for-age, BMI, and weight-for-length z-scores at 12 months than those fed standard formulas, and after controlling for other covariates, weight-for-age and BMI z-scores remained significantly higher in the non-standard formula infants. At 24 months, only weight-for-age z-scores remained significantly higher for non-standard formula users compared to standard formula users. Both non-standard and standard formula groups showed significantly higher weight-for-age, BMI, weight-for-length, and length-for-age z-scores compared to breastfed infants at 12 and 24 months. **Conclusions**: Non-standard infant formula may have a differential effect on growth during the first year of life compared to standard infant formula and breastfeeding. Future research should explore the long-term health effects of non-standard infant formula use and the risk of obesity.

## 1. Introduction

Childhood obesity affects nearly 20% of children and adolescents in the United States [1], with nutritional exposure during the first 1000 days of life playing a critical role in shaping metabolic programming, gut microbiome development, and long-term health outcomes [2,3,4,5]. Although breast milk remains the optimal source of infant nutrition [6], only about 23% of U.S. infants are exclusively breastfed at six months of age [7]. The American Academy of Pediatrics and Dietary Guidelines for Americans recommend iron-fortified formula for non-breastfed infants, but do not provide guidance on specific formula types [6,8]. Furthermore, although the U.S. Food and Drug Administration (FDA) mandates minimum levels for approximately 30 essential nutrients in infant formula, it provides limited regulatory guidance on the types or sources of macronutrients, such as proteins, fats, and carbohydrates, that may be included [9,10]. As a result, the infant formula market has seen a proliferation of non-standard products that differ from standard formulas by altering the type or amount of carbohydrate, protein, and fat.

A recent market analysis showed that nearly 59% of powdered formulas sold in major U.S. retail outlets are lactose-reduced or lactose-free, often substituting lactose with maltodextrin or sucrose and modifying protein content through partial or extensive hydrolysis [9]. These changes can increase the glycemic load, alter gut microbial colonization, and potentially accelerate weight gain [11]. Moreover, these non-standard formulas are frequently used in the absence of clinical indications, such as cow-milk protein allergy or congenital lactose intolerance, both of which are rare in infancy compared to the use of altered infant formulas [9]. While lactose in human milk supports the growth of beneficial gut bacteria like *Bifidobacteria*, which produce short-chain fatty acids essential for immune and metabolic health, lactose-free alternatives may disrupt this microbial balance [12,13]. Despite their widespread use, there is limited and inconsistent evidence on how non-standard formulas affect growth and metabolic outcomes [9,14]. Given rising obesity rates and the potential for early-life dietary exposure to influence long-term health, it is critical to evaluate how these formulas impact early childhood growth. This study addresses this gap by examining growth outcomes at one and two years among infants fed standard versus non-standard formulas.

## 2. Materials and Methods

### 2.1. Study Design

This retrospective cohort study evaluated the effects of different cow-milk-based infant formula types and breastfeeding on growth outcomes in infants and toddlers. The cohort included children under 3 years of age whose mothers received care within the University of Iowa Health Care system (UIHC) in the outpatient general pediatrics clinics between January 2017 and January 2024.

### 2.2. Population

Full-term infants with a birth weight of at least 2500 g formed the study population. Infants were included if they were followed for routine well-child visits and if growth parameter data were available at 12 and/or 24 months, with at least one time point for outcomes. Infants with a history of tube feeding, metabolic disorders, congenital anomalies, or seizure disorders were excluded, as these medical conditions can independently affect growth outcomes regardless of feeding practices [15,16,17,18].

### 2.3. Exposures

The primary exposure of interest was non-standard infant formula, which was defined as the use of infant formula with modifications in macronutrient composition, including altered protein or carbohydrate content, at 2 months of age. Examples include formulas with partially or extensively hydrolyzed proteins or formulas where lactose was reduced and replaced with other carbohydrates, such as maltodextrin or sucrose. We excluded children on soy-based or specialized formulas, such as those for metabolic disorders. The comparison group included standard cow-milk-based formula, which was defined as the use of infant formula containing intact cow-milk protein with lactose as the primary carbohydrate at 2 months of age. We also had a reference group of children with exclusive breastfeeding at 2 months of age. Children with mixed feeding (breastfeeding and formula feeding) were categorized as formula-fed and categorized based on the type of formula used (non-standard vs. standard infant formula).

### 2.4. Outcomes

Growth outcomes were assessed using z-scores calculated from WHO growth charts at 12 and 24 months of age [19]. These included:

Weight-for-age z-scores: An indicator of overall weight status relative to age.

Length-for-age z-scores: A measure of linear growth (stature) relative to age.

Weight-for-length z-scores: An indicator of body proportionality and a proxy for adiposity.

BMI-z-scores: A ratio of weight to height and considered a proxy for adiposity [20].

### 2.5. Data Collection

Data were collected from the electronic health records of children followed at the UIHC outpatient general pediatrics clinics between January 2017 and January 2024. The primary exposure and comparison data were obtained from caregiver responses to the Bright Futures Pre-Visit Questionnaire [21] administered at the 2-month well-child visit (Appendix A). Growth outcome data, including z-scores for weight-for-age, length-for-age, weight-for-length, and BMI, were calculated from actual weight and length measurements obtained during well-child visits at 1 and 2 years of age. Additional data on confounding variables, such as maternal health and socioeconomic factors, were extracted from linked maternal and infant medical records using the Intergenerational Health Knowledgebase (IHK) [22]. The database integrates information from all pregnancies receiving care at the UIHC, linking maternal and child data across episodes of care. It encompasses electronic health record (EHR) data, including maternal and child diagnoses, medications, vitals, hospital admissions, depression screenings, laboratory results, and procedures. Based on a review of the literature, several potential confounders were identified and included in the analyses [23,24,25,26]. These were:

Infant factors: Gestational age, birth weight, and sex.

Maternal factors: Maternal age, obesity, anemia, gestational diabetes, and hypertensive disorders during pregnancy.

Socioeconomic factors: Type of health insurance (Medicaid vs. private insurance) as a proxy for socioeconomic status.

### 2.6. Statistical Analysis

Summary statistics were calculated for demographic and clinical factors stratified by feeding group (breastfed, standard formula, and non-standard formula). Categorical measures are reported as counts and percentages. Continuous variable distributions were assessed for normality and reported as means and standard deviations or medians and interquartile ranges. Between-group homogeneity was assessed using Pearson’s chi-square and Kruskal–Wallis tests for categorical and continuous measures, respectively.

Generalized linear mixed models (GLMMs) were fit to evaluate unadjusted and adjusted associations between the type of infant formula (e.g., non-standard vs. standard), exclusive of breastfeeding growth outcomes. These models accounted for repeated measures on individual participants, and they were specified to handle the hierarchical structure of the data, ensuring appropriate adjustments for within-subject correlations. The models were adjusted for the following infant factors: gestational age, birth weight, sex; maternal factors: age, obesity, anemia, gestational diabetes, and hypertensive disorders during pregnancy; and socioeconomic factors: type of health insurance (Medicaid vs. private insurance) as a proxy for socioeconomic status. Mean differences with 95% confidence intervals and *p*-values were reported for between-group assessments of continuous outcomes. Relative risks with 95% confidence intervals and *p*-values were provided for between-group assessments of dichotomous outcomes. To account for multiple comparisons within each outcome measure, a Bonferroni correction was applied to the standard alpha = 0.05 cutoff, giving us alpha* = 0.05/6 = 0.0083 to achieve statistical significance. Additional subgroup analyses were performed to study the differential effect of non-standard infant formula based on gender, race, insurance type, and birth weight. All analyses were conducted using SAS version 9.4 (SAS Institute, Cary, NC, USA). This study was approved by the University of Iowa Institutional Review Board (IRB No. 202506109). The preliminary results were presented in Pediatric Academic Society meeting 2025 [27].

## 3. Results

From an initial sample of 11,047 infants, 5532 were excluded, resulting in a final cohort of 5515 infants who met the eligibility criteria and were included in the final analysis (Figure 1). We compared the baseline characteristics of included infants (n = 5515) and those excluded due to missing data (n = 4331). The two groups were broadly similar in terms of sex, race, and maternal race. Small but statistically significant differences were observed for insurance type and maternal ethnicity, with the included group having a slightly higher proportion of private insurance and Hispanic mothers. These comparisons are presented in Appendix A.

The baseline demographic and clinical characteristics of the study population are summarized in Table 1. At 2 months, feeding practices showed 35% breastfeeding, 42% standard formula, and 23% non-standard formula. Sex distribution was similar across feeding groups. Notable differences included higher rates of private insurance and mothers identified as White race, as well as lower maternal obesity and anemia, in the breastfeeding group. Breastfed infants also had slightly higher gestational age and birth weight (likely driven by large sample size) and were more likely to continue breastmilk use at one year, while formula and cow’s milk remained more common in the formula-fed groups.

Infants consuming non-standard infant formulas had higher weight-for-age z-scores (adjusted mean difference 0.146, 95% CI, 0.086–0.206) and higher BMI z-scores (adjusted mean difference 0.122, 95% CI, 0.041–0.203) at 12 months compared to the standard formula group. A significantly greater weight-for-length z-score was observed in the non-standard formula compared to the standard formula at 12 months, which became nonsignificant after adjusting for covariates. No significant differences between feeding groups were observed in length-for-age z-scores. At 24 months, only the weight-for-age z-score was significantly greater in the non-standard formula group compared to the standard formula, which persisted after confounder adjustment (adjusted mean difference 0.093, 95% CI, 0.020, 0.166), while BMI, length-for-age, and weight-for-length z-scores remained nonsignificant (Table 2 and Figure 2).

Infants who consumed non-standard and standard formulas consistently showed significantly higher growth z-scores compared to breastfed infants across weight-for-age, BMI, length-for-age, and weight-for-length outcomes at both 12 and 24 months. These differences persisted even after adjusting for potential confounders (Figure 2).

### Subgroup Analyses

Among females, non-standard formula use was associated with significantly higher BMI z-scores at 12 months of age (mean difference (MD) = 0.229, 95% CI, 0.16, 0.332) compared to the standard formula (Table 3). No such significant difference was seen for males (MD 0.090, 95% CI −0.018–0.198) (Table 3). Race-based analyses showed significant differences for White infants at 12 months, with non-standard formula linked to higher BMI z-scores compared to standard formula (effect size = 0.164, 95% CI, 0.067, 0.260). Ethnicity-based results revealed a significant difference among non-Hispanic infants, where non-standard formula was associated with higher BMI z-scores at 12 months compared to standard formula, although the magnitude of effect was higher for the Hispanic population in both standard and non-standard infant formula compared to breastfeeding (Table 2). For insurance type, infants with Medicaid or private insurance showed significantly higher BMI z-scores with the non-standard formula at 12 months. Similarly, among infants with birth weight >4000 g, the non-standard formula was associated with a higher BMI z-score effect size at 12 months compared to the standard formula, but it was not statistically significant (MD 0.261, 95% CI −0.028, 0.55).

## 4. Discussion

In this large retrospective cohort study of over 5500 full-term infants, we found that non-standard infant formulas, defined by alterations in protein or carbohydrate content, were associated with higher weight-for-age and BMI z-scores at 12 months compared to standard formulas. While weight-for-age z-scores remained significantly elevated at 24 months among non-standard formula users, differences in BMI and weight-for-length diminished after adjustment for confounders. Importantly, both standard and non-standard formula groups demonstrated consistently higher growth z-scores across weight-for-age, BMI, weight-for-length, and length-for-age measures compared to breastfed infants at both 12 and 24 months. Subgroup analyses suggested that the impact of non-standard formula may be more pronounced in specific populations, including females, White infants, those with private or Medicaid insurance, and those with higher birth weight. These findings raise important questions about the metabolic and growth-related consequences of altered formula composition during infancy. After applying a Bonferroni correction for multiple comparisons, fewer associations remained statistically significant; however, the overall patterns of higher growth z-scores among formula-fed infants compared to breastfed infants were consistent.

Our results align with previous studies that associate non-standard or altered formulas with increased weight gain and higher BMI z-scores. Young et al. found that sucrose-based infant formulas are associated with higher weight-for-length and abdominal circumference z-scores in infants aged 0–12 months, potentially predisposing them to excessive weight gain and increased obesity risk [28]. In our study, we included all the non-standard infant formulas that could have included added carbohydrates such as maltodextrin, sucrose and others. Merritt et al. raised concerns about these formulas’ metabolic and microbiome effects, noting potential links to obesity and altered infant gut development [29]. The underlying mechanism for differential growth could be potentially linked to differences in the absorption of macronutrients and their effect on the microbiome. For example, the glucose polymers such as maltodextrins, commonly used in lactose-free formulas, are hydrolyzed by α-amylase into glucose, resulting in rapid absorption and higher glycemic responses with a glycemic index (GI) of 110, compared to lactose, GI of 46 [30]. These substituted sugars may disrupt glycemic regulation, enhance sweet taste preference, and reduce satiety signaling, all of which can promote overfeeding and rapid weight gain, leading to a higher risk of childhood obesity [14]. Similarly, hydrolyzed protein formulas may alter metabolic responses due to faster absorption rates compared to intact proteins [31]. While our study found significant differences in weight-for-age and BMI z-scores, the lack of difference in length-for-age z-scores suggests that altered formula composition may primarily affect adiposity rather than linear growth. We did not include categorical outcomes of overweight and obesity in our analysis because such definitions are less well established in infants and toddlers under 2 years, and current WHO and CDC recommendations emphasize the use of continuous growth z-scores in this age group. Continuous measures also provide greater statistical power to detect subtle differences in growth patterns. Nonetheless, the elevated weight-for-age and BMI z-scores observed among infants fed non-standard formulas may reflect growth trajectories associated with increased risk of overweight and obesity later in childhood, a question that warrants prospective follow-up.

This study has several strengths. It includes a large sample size of 5515 infants, enabling robust comparisons across feeding groups and demographic subgroups. Our study contributes novel insights to the literature on infant feeding and growth in several ways. First, unlike prior investigations that have focused on single formula subtypes or small trial populations [11,32,33], we evaluated a large, real-world cohort of over 5500 infants using electronic health record data, which enhances relevance to clinical practice. Second, we examined a broad spectrum of non-standard formulas, including lactose-reduced and partially hydrolyzed products that account for the majority of non-standard formula use in the United States [9], thereby reflecting the feeding patterns most relevant to families and clinicians. Third, our comparative design simultaneously assessed exclusive breastfeeding, standard formula, and non-standard formula, providing a comprehensive framework for understanding differences across feeding modalities. Finally, we assessed growth outcomes at both 12 and 24 months, extending beyond the short-term outcomes typically required for formula approval studies and offering a more longitudinal perspective on how early nutritional exposures may shape growth trajectories. Together, these features underscore the novelty of our study and its contribution to the evidence base guiding infant feeding decisions. Also, the comprehensive adjustment for confounding variables, including maternal characteristics, birth weight, and socioeconomic factors, strengthens the association inferences. Lastly, the inclusion of subgroup analyses provides detailed insights into how sex, race, ethnicity, insurance type, and birth weight may influence growth outcomes, offering valuable information for future research.

However, despite these strengths, this study has several limitations. A key limitation of this study is that infant feeding exposure was captured only at the 2-month well-child visit, without follow-up data on formula or breastfeeding practices beyond that timepoint. We selected the 2-month mark intentionally, as it represents a critical inflection point in infant feeding in the United States. Although many mothers initiate breastfeeding, the most significant decline occurs within the first two months postpartum, often coinciding with the end of maternity leave and return to work. Thus, feeding practices at 2 months may be a reasonable proxy for sustained nutritional exposure throughout the remainder of infancy [34]. Nonetheless, this approach does not account for early feeding exposures, changes in formula type or feeding modality after 2 months, or mixed feeding dynamics over time. Future prospective studies with longitudinal feeding data are needed to capture these evolving patterns and more accurately assess cumulative nutritional impact.

The retrospective design relies on routinely collected electronic health record data, which may be subject to documentation inconsistencies and missing information. While we adjusted for potential confounders, residual confounding from unmeasured variables cannot be ruled out. For example, we adjusted for a number of maternal, infant, and socioeconomic covariates; we were unable to include other potentially important sociodemographic and psychosocial variables such as maternal education, occupation, or family support, as these data were not consistently available in our dataset. Maternal depression screening results were partially available, but incomplete reporting limited their inclusion. We therefore used health insurance type (Medicaid vs. private) as a proxy for socioeconomic status, an approach consistent with prior pediatric studies with the acknowledgment of its limitations to represent the socioeconomic status [35,36]. Future research should incorporate more granular measures of maternal education, mental health, and psychosocial context to better characterize their influence on infant feeding choices and growth outcomes.

Another limitation is that by 12 months of age, infants’ diets typically diversify, and dietary intake between ages 1 and 2 years may have a stronger impact on growth at 24 months than feeding type at 2 months. This dietary transition likely attenuated the associations we observed at 24 months. Nonetheless, reporting these outcomes remains important, as early nutritional exposures can influence growth trajectories beyond infancy, and we found that weight-for-age differences persisted into the second year of life

Our cohort may not fully represent the broader U.S. population, limiting generalizability. The amount of infant formula, caloric density, and reasons for using non-standard formula were not reported in the pre-visit questionnaire, all of which can significantly impact growth. Moreover, we categorized multiple types of infant formulas into a single category of non-standard formula. This grouping was necessary because the number of infants receiving extensively hydrolyzed or lactose-free formulas was very small, limiting the feasibility of stratified analyses. Most non-standard formulas in our cohort were lactose-reduced and partially hydrolyzed products, which recent analyses have shown to share broadly similar macronutrient profiles, including replacement of lactose with added sugars such as maltodextrin, sucrose, or glucose [14]. We acknowledge that fully hydrolyzed and lactose-free formulas may have differential effects on growth, and future studies with larger sample sizes are needed to disentangle these subtype-specific associations.

Subgroup analyses suggested that the effects of non-standard versus standard formulas may differ by sex, race/ethnicity, and insurance type, with some differences attenuating in specific groups (e.g., Black infants). These findings underscore the importance of considering sociodemographic and biological context when evaluating infant growth outcomes. However, given the reduced sample sizes in subgroup analyses, these results should be interpreted cautiously, and larger, more diverse cohorts are needed to confirm and expand upon these observations.

Although all infant formulas sold in the U.S. must meet FDA safety standards, current regulatory requirements are limited to demonstrating adequate infant growth over a short period, typically 15 weeks [37]. There is no requirement to assess longer-term outcomes such as weight gain trajectories, adiposity, metabolic programming, or neurodevelopment, factors that are increasingly recognized as critical determinants of lifelong health [37,38]. Given our findings that non-standard formulas may influence early weight gain and adiposity, future research should go beyond short-term adequacy and rigorously evaluate the long-term metabolic and developmental impacts of altered formula composition. Prospective cohort studies are needed to assess the comparative effects of different formula types, including partially hydrolyzed, extensively hydrolyzed, lactose-reduced, and lactose-free formulations, on obesity risk, insulin sensitivity, gut microbiome composition, and cognitive outcomes. Future studies should also incorporate detailed data on feeding patterns, volume, and duration of formula use, and the underlying reasons for formula selection, including caregiver and provider decision-making. Additionally, research should address how marketing and labeling influence the use of non-standard formulas in infants without medical indications. Such data are essential to inform regulatory standards and clinical guidelines that prioritize not only short-term growth but also long-term health outcomes.

## 5. Conclusions

Non-standard infant formulas appear to have a differential impact on early growth compared to standard formulas and breastfeeding, particularly in the first year of life. Our findings highlight the need for clearer clinical guidance regarding their use in infants without medical indications. As current guidelines do not distinguish between formula types, further research is essential to evaluate the long-term metabolic and developmental consequences of non-standard formulas and to better understand the factors influencing their widespread use.

## Figures and Tables

**Figure 1 nutrients-17-03111-f001:**
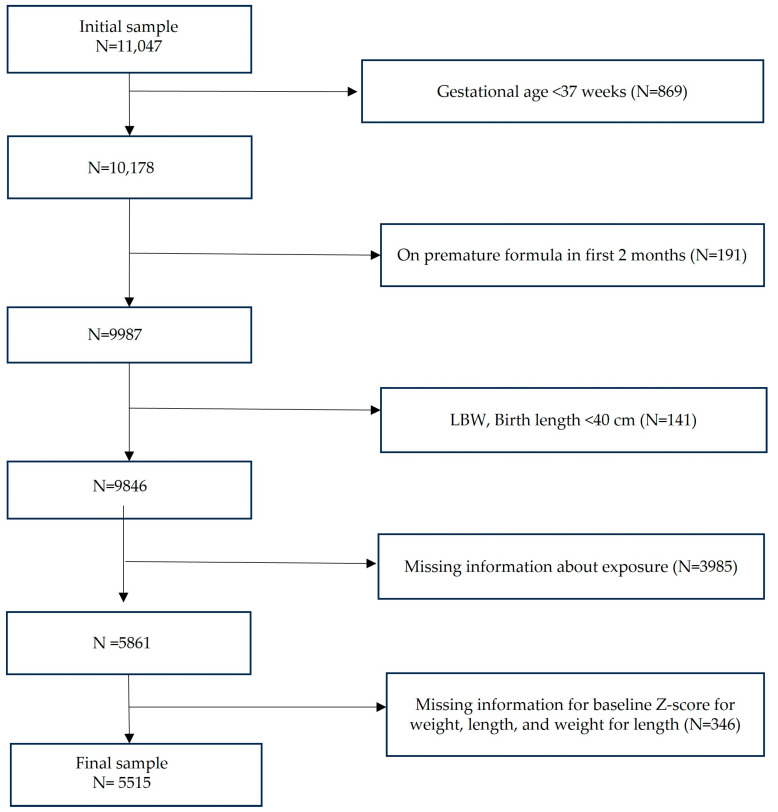
Flow diagram of sample size; LBW = Low Birth Weight (<2500 g). The diagram shows the inclusion of participants in the final analysis. The data were extracted from electronic medical records.

**Figure 2 nutrients-17-03111-f002:**
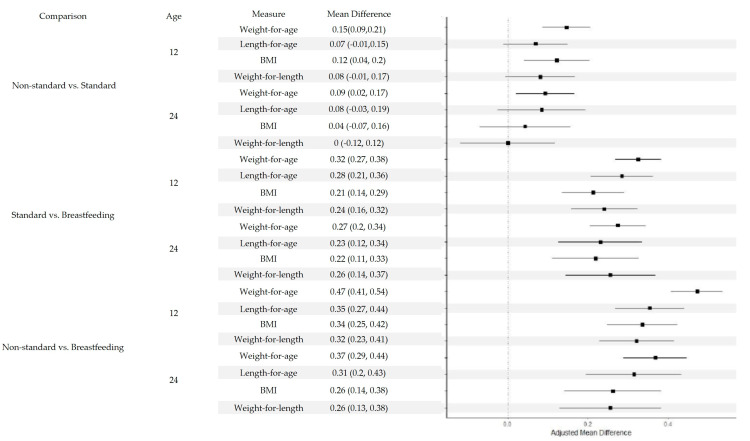
Adjusted Mean Differences in Growth Z-scores by Feeding Type at 12 and 24 Months. The scores are WHO-based z-scores. The analyses were adjusted for infant and maternal confounders: infant Gestational age, birth weight, and sex; Maternal age, obesity, anemia, gestational diabetes, and hypertensive disorders during pregnancy; and Socioeconomic factors: Type of health insurance, Abbreviation: BMI: Body Mass Index.

**Table 1 nutrients-17-03111-t001:** Demographics and Clinical Characteristics.

Variable	Breastfeeding (N = 1922)	Standard Formula (N = 2301)	Non-Standard Formula (N = 1292)	*p*-Value
Sex, N (%)				0.2763
Female	979 (50.9%)	1143 (49.7%)	621 (48%)	
Male	944 (49.1%)	1158 (50.3%)	672 (52%)	
Insurance, N (%)				<0.0001
Medicaid	355 (18.8%)	816 (35.8%)	421 (32.9%)	
Other	374 (19.9%)	482 (21.2%)	275 (21.5%)	
Private	1155 (61.3%)	981 (43.1%)	584 (45.6%)	
Race, N (%)				<0.0001
Black	83 (5.1%)	407 (20.3%)	121 (11.4%)	
Other (Asian, American Indians, Pacific Islanders)	237 (14.6%)	482 (24%)	196 (18.5%)	
White	1309 (80.4%)	1116 (55.7%)	741 (70%)	
Gestational Diabetes, N (%)	153 (9.4%)	224 (11.2%)	112 (10.6%)	0.2134
Maternal Obesity, N (%)	217 (13.3%)	409 (20.4%)	265 (25.1%)	<0.0001
Maternal Anemia, N (%)	372 (22.8%)	602 (30%)	282 (26.7%)	<0.0001
Mother’s Age (Years), Median (IQR)	31 (28–34)	30 (26–34)	30 (26–33)	<0.0001
Gestational Age (Days), Median (IQR)	276 (272–281)	275 (271–280)	274 (270–279)	<0.0001
Birth Weight (g), Median (IQR)	3445 (3165–3748)	3375 (3095–3685)	3385 (3095–3680)	<0.0001
Birth Length (cm), Median (IQR)	51.0 (49.5–52.5)	51 (49.5–52.1)	50.8 (49.5–52.1)	0.0035
Cow’s milk at 1 year of age *	422 (47.0%)	850 (66.3%)	490 (63.6%)	<0.0001
Breast milk at 1 year of age *	651 (72.5%)	236 (18.4%)	86 (11.2%)	<0.0001
Formula at 1 year of age *	52 (5.8%)	524 (40.8%)	347 (45.1%)	<0.0001
Soy milk at 1 year of age *	16 (1.8%)	13 (1.0%)	14 (1.8%)	0.2103

* Percentages for milk types at 1 year (breast milk, cow’s milk, formula, soy milk) do not add up to 100% because families could report more than one milk type at the 1-year visit (e.g., both breast milk and cow’s milk).

**Table 2 nutrients-17-03111-t002:** Comparison of unadjusted and adjusted growth Z-Scores across feeding groups at 12 and 24 Months of age.

Feeding Group Comparison	Age (Months)	Z-Score	Unadjusted Mean Diff (95% CI)	Unadjusted *p*-Value	Adjusted Mean Difference (95% CI)	*p*-Value
Non-standard vs. Standard	12	Weight-for-age	0.154 (0.094, 0.214)	<0.0001	0.146 (0.086, 0.206)	<0.0001
	12	Length-for-age	0.034 (0.046, 0.114)	0.4055	0.069 (−0.011, 0.149)	0.0898
	12	BMI	0.146 (0.065, 0.228)	0.0004	0.122 (0.041, 0.203)	0.0032
	12	Weight-for-length	0.088 (0.001, 0.176)	0.0487	0.081 (−0.006, 0.167)	0.0681
	24	Weight-for-age	0.103 (0.031, 0.176)	0.0052	0.093 (0.020, 0.166)	0.012
	24	Length-for-age	0.045(−0.065, 0.156)	0.4209	0.084 (−0.026, 0.194)	0.1356
	24	BMI	0.068 (−0.045, 0.181)	0.2368	0.043 (−0.070, 0.156)	0.4552
	24	Weight-for-length	0.010 (−0.109, 0.129)	0.8714	0.000 (−0.118, 0.118)	0.9997
Standard vs. Breastfeeding	12	Weight-for-age	0.339 (0.283, 0.395)	<0.0001	0.324 (0.267, 0.382)	<0.0001
	12	Length-for-age	0.334 (0.258, 0.410)	<0.0001	0.284 (0.207, 0.361)	<0.0001
	12	BMI	0.207 (0.130, 0.284)	<0.0001	0.212 (0.135, 0.290)	<0.0001
	12	Weight-for-length	0.235 (0.153, 0.318)	<0.0001	0.240 (0.157, 0.323)	<0.0001
	24	Weight-for-age	0.295 (0.226, 0.363)	<0.0001	0.274 (0.204, 0.343)	<0.0001
	24	Length-for-age	0.284 (0.179, 0.388)	<0.0001	0.230 (0.125, 0.335)	<0.0001
	24	BMI	0.212 (0.106, 0.319)	<0.0001	0.218 (0.111, 0.326)	<0.0001
	24	Weight-for-length	0.249 (0.137, 0.361)	<0.0001	0.255 (0.144, 0.367)	<0.0001
Non-standard vs. Breastfeeding	12	Weight-for-age	0.493 (0.429, 0.557)	<0.0001	0.471 (0.406, 0.535)	<0.0001
	12	Length-for-age	0.368 (0.282, 0.454)	<0.0001	0.353 (0.267, 0.439)	<0.0001
	12	BMI	0.354 (0.266, 0.441)	<0.0001	0.335 (0.247, 0.422)	<0.0001
	12	Weight-for-length	0.324 (0.230, 0.418)	<0.0001	0.320 (0.227, 0.414)	<0.0001
	24	Weight-for-age	0.398 (0.320, 0.476)	<0.0001	0.367 (0.288, 0.445)	<0.0001
	24	Length-for-age	0.329 (0.211, 0.447)	<0.0001	0.314 (0.195, 0.432)	<0.0001
	24	BMI	0.280 (0.160, 0.401)	<0.0001	0.262 (0.141, 0.382)	<0.0001
	24	Weight-for-length	0.259 (0.132, 0.385)	<0.0001	0.255 (0.1290.382)	<0.0001

**Table 3 nutrients-17-03111-t003:** Subgroup analyses summarizing the mean difference in unadjusted BMI z-scores across feeding types at 12 and 24 months.

				Unadjusted			
Subset	Age (Months)	Feeding	Reference	Mean Diff	95% Lower	95% Upper	* p * -Value
Male	12	Non-standard Formula	Standard Formula	0.090	−0.018	0.198	0.1036
	12	Standard Formula	Breastfeeding	0.232	0.128	0.336	<0.0001
	12	Non-standard Formula	Breastfeeding	0.322	0.206	0.438	<0.0001
	24	Non-standard Formula	Standard Formula	0.066	−0.086	0.217	0.396
	24	Standard Formula	Breastfeeding	0.204	0.061	0.346	0.0051
	24	Non-standard Formula	Breastfeeding	0.269	0.108	0.431	0.0011
Female	12	Non-standard Formula	Standard Formula	0.229	0.126	0.332	<0.0001
	12	Standard Formula	Breastfeeding	0.194	0.098	0.291	<0.0001
	12	Non-standard Formula	Breastfeeding	0.424	0.314	0.533	<0.0001
	24	Non-standard Formula	Standard Formula	0.079	−0.065	0.224	0.2824
	24	Standard Formula	Breastfeeding	0.255	0.116	0.394	0.0003
	24	Non-standard Formula	Breastfeeding	0.334	0.179	0.489	<0.0001
Black	12	Non-standard Formula	Standard Formula	0.035	−0.216	0.287	0.7823
	12	Standard Formula	Breastfeeding	0.202	−0.117	0.521	0.2141
	12	Non-standard Formula	Breastfeeding	0.237	−0.127	0.602	0.2022
	24	Non-standard Formula	Standard Formula	−0.209	−0.552	0.134	0.2319
	24	Standard Formula	Breastfeeding	0.429	−0.048	0.906	0.0784
	24	Non-standard Formula	Breastfeeding	0.220	−0.304	0.743	0.4108
White	12	Non-standard Formula	Standard Formula	0.164	0.067	0.260	0.0009
	12	Standard Formula	Breastfeeding	0.232	0.143	0.321	<0.0001
	12	Non-standard Formula	Breastfeeding	0.396	0.299	0.493	<0.0001
	24	Non-standard Formula	Standard Formula	0.101	−0.035	0.236	0.1464
	24	Standard Formula	Breastfeeding	0.195	0.073	0.317	0.0018
	24	Non-standard Formula	Breastfeeding	0.295	0.160	0.431	<0.0001
Other	12	Non-standard Formula	Standard Formula	−0.028	−0.214	0.158	0.7687
	12	Standard Formula	Breastfeeding	0.417	0.228	0.607	<0.0001
	12	Non-standard Formula	Breastfeeding	0.389	0.168	0.611	0.0006
	24	Non-standard Formula	Standard Formula	0.051	−0.197	0.299	0.6889
	24	Standard Formula	Breastfeeding	0.424	0.160	0.688	0.0017
	24	Non-standard Formula	Breastfeeding	0.474	0.174	0.775	0.002
Hispanic	12	Non-standard Formula	Standard Formula	−0.030	−0.279	0.219	0.8134
	12	Standard Formula	Breastfeeding	0.476	0.178	0.774	0.0018
	12	Non-standard Formula	Breastfeeding	0.446	0.117	0.776	0.008
	24	Non-standard Formula	Standard Formula	0.069	−0.253	0.390	0.676
	24	Standard Formula	Breastfeeding	0.204	−0.204	0.612	0.3272
	24	Non-standard Formula	Breastfeeding	0.273	−0.162	0.707	0.2193
Non-Hispanic	12	Non-standard Formula	Standard Formula	0.170	0.084	0.256	0.0001
	12	Standard Formula	Breastfeeding	0.180	0.099	0.260	<0.0001
	12	Non-standard Formula	Breastfeeding	0.349	0.258	0.440	<0.0001
	24	Non-standard Formula	Standard Formula	0.080	−0.041	0.202	0.1963
	24	Standard Formula	Breastfeeding	0.195	0.084	0.307	0.0006
	24	Non-standard Formula	Breastfeeding	0.276	0.148	0.403	<0.0001
Medicaid	12	Non-standard Formula	Standard Formula	0.179	0.040	0.318	0.0116
	12	Standard Formula	Breastfeeding	0.315	0.150	0.481	0.0002
	12	Non-standard Formula	Breastfeeding	0.494	0.313	0.676	<0.0001
	24	Non-standard Formula	Standard Formula	0.042	−0.154	0.237	0.676
	24	Standard Formula	Breastfeeding	0.269	0.034	0.505	0.025
	24	Non-standard Formula	Breastfeeding	0.311	0.058	0.564	0.0162
Private	12	Non-standard Formula	Standard Formula	0.203	0.099	0.307	0.0001
	12	Standard Formula	Breastfeeding	0.173	0.082	0.265	0.0002
	12	Non-standard Formula	Breastfeeding	0.376	0.273	0.480	<0.0001
	24	Non-standard Formula	Standard Formula	0.058	−0.088	0.204	0.4348
	24	Standard Formula	Breastfeeding	0.213	0.087	0.339	0.0009
	24	Non-standard Formula	Breastfeeding	0.271	0.125	0.417	0.0003
Other insurance	12	Non-standard Formula	Standard Formula	0.028	−0.138	0.194	0.7426
	12	Standard Formula	Breastfeeding	0.274	0.112	0.436	0.0009
	12	Non-standard Formula	Breastfeeding	0.302	0.124	0.480	0.0009
	24	Non-standard Formula	Standard Formula	0.156	−0.077	0.388	0.1891
	24	Standard Formula	Breastfeeding	0.173	−0.059	0.405	0.1434
	24	Non-standard Formula	Breastfeeding	0.329	0.074	0.584	0.0114
<4000 g	12	Non-standard Formula	Standard Formula	0.140	0.055	0.224	0.0012
	12	Standard Formula	Breastfeeding	0.189	0.109	0.270	<0.0001
	12	Non-standard Formula	Breastfeeding	0.329	0.239	0.420	<0.0001
	24	Non-standard Formula	Standard Formula	0.087	−0.032	0.205	0.1518
	24	Standard Formula	Breastfeeding	0.178	0.066	0.290	0.0018
	24	Non-standard Formula	Breastfeeding	0.265	0.139	0.390	<0.0001
>4000 g	12	Non-standard Formula	Standard Formula	0.261	−0.028	0.550	0.0765
	12	Standard Formula	Breastfeeding	0.375	0.129	0.621	0.0028
	12	Non-standard Formula	Breastfeeding	0.636	0.331	0.941	<0.0001
	24	Non-standard Formula	Standard Formula	−0.034	−0.412	0.344	0.8587
	24	Standard Formula	Breastfeeding	0.537	0.193	0.881	0.0023
	24	Non-standard Formula	Breastfeeding	0.503	0.087	0.918	0.0179

Other race = American Indian, Alaska Native, Asian, Native Hawaiian.

## Data Availability

The data for this study is available from corresponding author and could be available on request due to technical/time limitations.

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
