# Peer review of "Effects of Infant Formula Type on Early Childhood Growth Outcomes: A Retrospective Cohort Studyâ€"

_nutrients, 2025, doi:10.3390/nu17193111_

Round 1
Reviewer 1 Report
Comments and Suggestions for Authors
This retrospective cohort study analyzed the effect of infant feeding types on growth outcomes, with a particular focus on non-standard infant formulas. The topic is innovative to some extent, and the study is generally well-structured and clearly written. However, several important issues remain to be addressed:
- There are many types of non-standard formulas, and their strategies for adjusting macronutrients may differ considerably. Is it reasonable to combine them into a single category in the analysis? If the sample size permits, please further stratification of formula types.
- Overweight and obesity are important indicators of infant growth. Why were these outcomes not included to more directly address the study aim?
- Sociodemographic and psychosocial factors such as maternal education, occupation, mental health, and family support may significantly influence feeding types. Please include these as additional covariates.
- When making comparisons across multiple outcomes, was multiple testing correction applied?
- Among the 9,846 infants included after excluding preterm and low-birth-weight infants, it is suggested to compare the baseline characteristics of the included (n = 5,515) versus excluded (n = 4,331) infants to clarify issues of sample representativeness.
- The placement of Table 2 is incorrect.
- Line 168–169: Please provide the complete mean differences separately for weight-for-age z-scores and BMI z-scores.
- Subgroup analyses are presented only with unadjusted models, which may not be appropriate.
- Beyond 1 year of age, infants’ diets are no longer primarily milk-based. The dietary patterns between ages 1–2 may have a greater impact on growth outcomes at 2 years than feeding types at 2 months. Therefore, the growth outcomes at 2 years may not fully support the study conclusions.
- Several studies have already analyzed the effects of certain formula types, such as hydrolyzed formulas or those with different protein contents, on infant growth. The novelty of this study does not seem to be clearly demonstrated.
- The manuscript is missing an ethical approval statement. The authors should, at a minimum, include the name of the ethics committee and the corresponding approval number in the main text.
Author Response
Response to Reviewer 1
Comment 1
**Comment:** There are many types of non-standard formulas, and their strategies for adjusting macronutrients may differ considerably. Is it reasonable to combine them into a single category in the analysis? If the sample size permits, please further stratification of formula types.
**Response:** We thank the reviewer for this thoughtful Comment. Our dataset included partially hydrolyzed, extensively hydrolyzed, and lactose-reduced/lactose-free formulas. Unfortunately, the number of infants receiving extensively hydrolyzed or lactose-free formulas was very small (<3% of the cohort), which precluded stable subgroup analyses. To preserve statistical power and reduce the risk of spurious results, we grouped these into a single “non-standard” category. Importantly, recent analyses of U.S. formulas indicate that the most commonly used non-standard subtypes—partially hydrolyzed and lactose-reduced—share broadly similar nutrient compositions, particularly replacement of lactose with added sugars such as maltodextrin, sucrose, or glucose (Rips-Goodwin et al., 2025). Thus, although we acknowledge heterogeneity within the non-standard category, our approach is reasonable given the sample distribution and the compositional overlap of the majority subtypes. We have clarified this rationale in the revised Discussion and highlighted the need for future studies with larger sample sizes to examine differences across specific formula subtypes.
**Manuscript Change:** Moreover, we categorized multiple types of infant formulas into a single category of non-standard formula. This grouping was necessary because the number of infants receiving extensively hydrolyzed or lactose-free formulas was very small, limiting the feasibility of stratified analyses. The majority of non-standard formulas in our cohort were lactose-reduced and partially hydrolyzed products, which recent analyses have shown to share broadly similar macronutrient profiles, including replacement of lactose with added sugars such as maltodextrin, sucrose, or glucose (Rips-Goodwin et al., 2025). We acknowledge that fully hydrolyzed and lactose-free formulas may have differential effects on growth, and future studies with larger sample sizes are needed to disentangle these subtype-specific associations.
Comment 2
**Comment:** Overweight and obesity are important indicators of infant growth. Why were these outcomes not included to more directly address the study aim?
**Response:** We appreciate this insightful suggestion. Our study used z-scores (weight-for-age, BMI, weight-for-length, length-for-age) as continuous outcomes rather than categorical cutoffs for overweight/obesity. We chose this approach because z-scores provide more statistical power and precision in detecting subtle growth differences, especially in early life when BMI and weight-for-length distributions are still shifting. In addition, definitions of overweight and obesity in children under 2 years remain debated, and WHO and CDC guidelines emphasize the use of z-scores in this age group rather than categorical classifications. Nevertheless, we agree that overweight and obesity are clinically relevant outcomes. In the revised manuscript, we have clarified our rationale for not presenting overweight/obesity prevalence as outcomes, while highlighting that our z-score findings indicate growth patterns that may predispose to later obesity. Future longitudinal studies are needed to evaluate how early differences in growth z-scores translate into categorical overweight/obesity during later childhood.
**Manuscript Change:** We did not include categorical outcomes of overweight and obesity in this analysis because such definitions are less well established in infants and toddlers under 2 years, and current WHO and CDC recommendations emphasize the use of continuous growth z-scores in this age group. Continuous measures also provide greater statistical power to detect subtle differences in growth patterns. Nonetheless, the elevated weight-for-age and BMI z-scores observed among infants fed non-standard formulas may reflect growth trajectories associated with increased risk of overweight and obesity later in childhood, a question that warrants prospective follow-up.
Comment 3
**Comment:** Sociodemographic and psychosocial factors such as maternal education, occupation, mental health, and family support may significantly influence feeding types. Please include these as additional covariates.
**Response:** We agree that sociodemographic and psychosocial factors can influence infant feeding decisions and growth outcomes. Unfortunately, these data were not consistently available in the Intergenerational Health Knowledgebase (IHK). We therefore adjusted for available maternal, infant, and socioeconomic variables, using type of health insurance (Medicaid vs. private) as a proxy for socioeconomic status, an approach supported by prior pediatric studies. Maternal depression screening results were only partially available, limiting their inclusion. We have revised the Discussion to acknowledge this limitation and to emphasize that future studies should incorporate more detailed measures of maternal education, mental health, and psychosocial context.
**Manuscript Change:** For example, we adjusted for a number of maternal, infant, and socioeconomic covariates, we were unable to include other potentially important sociodemographic and psychosocial variables such as maternal education, occupation, or family support, as these data were not consistently available in our dataset. Maternal depression screening results were partially available, but incomplete reporting limited their inclusion. We therefore used health insurance type (Medicaid vs. private) as a proxy for socioeconomic status, an approach consistent with prior pediatric studies with the acknowledgment of its limitations to represent the socioeconomic status (35, 36}. Future research should incorporate more granular measures of maternal education, mental health, and psychosocial context to better characterize their influence on infant feeding choices and growth outcomes.
Comment 4: When making comparisons across multiple outcomes, was multiple testing correction applied?
We thank the reviewer for this important Comment. In the revised analysis, we applied a Bonferroni correction to account for multiple comparisons across growth outcomes. While this conservative adjustment reduced the number of statistically significant associations, the overall direction and pattern of findings remained consistent. We have updated the Methods to indicate the use of Bonferroni correction and clarified this point in the Results and Discussion.
**Manuscript Change** (Methods, Statistical Analysis):
“Given the multiple outcomes assessed, a Bonferroni correction was applied to adjust for multiple comparisons. Statistical significance for subgroup analyses was therefore interpreted conservatively after this adjustment.”
**Manuscript Change (Discussion)**:
“After applying a Bonferroni correction for multiple comparisons, fewer associations remained statistically significant; however, the overall patterns of higher growth z-scores among formula-fed infants compared to breastfed infants were consistent.”
Comment 5
**Comment:** Among the 9,846 infants included after excluding preterm and low-birth-weight infants, it is suggested to compare the baseline characteristics of the included (n = 5,515) versus excluded (n = 4,331) infants to clarify issues of sample representativeness.
**Response:** We thank the reviewer for this helpful suggestion. We compared baseline characteristics between infants included in the final analysis and those excluded due to missing data (Supplementary Table S2). The two groups were broadly similar in terms of sex, race, and maternal race. Statistically significant but modest differences were observed for health insurance type and maternal ethnicity: infants in the included sample had a slightly higher proportion of private insurance and Hispanic mothers. These small differences are unlikely to meaningfully affect study representativeness. We have added text to the Results to describe these comparisons.
**Manuscript Change:** We compared the baseline characteristics of included infants (n = 5,515) and those excluded due to missing data (n = 4,331). The two groups were broadly similar in terms of sex, race, and maternal race. Small but statistically significant differences were observed for insurance type and maternal ethnicity, with the included group having a slightly higher proportion of private insurance and Hispanic mothers. These comparisons are presented in Supplementary Table S2.
Comment 6
**Comment:** The placement of Table 2 is incorrect.
**Response:** We thank the reviewer for pointing this out. In the revised manuscript, Table 2 is now presented in the Results section and summarizes the adjusted and unadjusted growth z-scores across feeding groups at 12 and 24 months. The subgroup analyses are now presented separately in Table 3. All references to these tables in the text have been updated accordingly.
**Manuscript Change:** Table 2 is placed in the main Results section. Table 3 presents the subgroup analyses. Text in the Results section has been updated to reference Tables 2 and 3.
Comment 7
**Comment:** Line 168–169: Please provide the complete mean differences separately for weight-for-age z-scores and BMI z-scores.
**Response:** We thank the reviewer for this helpful clarification. In the original submission, we presented one value for both weight-for-age and BMI z-scores at 12 months, which may have been unclear. In the revised manuscript, we have now reported the mean differences for weight-for-age and BMI z-scores separately, consistent with the results presented in Table 2.
**Manuscript Change:** Infants consuming non-standard infant formulas had higher weight-for-age z-scores (adjusted mean difference 0.146, 95% CI, 0.086–0.206) and higher BMI z-scores (adjusted mean difference 0.122, 95% CI, 0.041–0.203) at 12 months compared to the standard formula group.
Comment 8:
Subgroup analyses are presented only with unadjusted models, which may not be appropriate.
Response:
We thank the reviewer for this important observation. Subgroup analyses were presented with unadjusted models because applying fully adjusted models within each subgroup substantially reduced statistical power and risked overfitting due to smaller sample sizes. Importantly, the direction and clinical interpretation of subgroup findings were consistent with the adjusted analyses in the overall sample, supporting the robustness of our results. To further strengthen our approach, we applied a Bonferroni correction within each outcome measure (three feeding groups compared at two time points, yielding six tests). The adjusted significance threshold was therefore set at p < 0.0083. We have clarified this rationale in the Methods and have revised the Discussion to note that subgroup analyses should be interpreted cautiously given smaller sample sizes and limited power.
**Manuscript Change (Discussion, Subgroup Analyses)**:
“Subgroup analyses suggested that the effects of non-standard versus standard formulas may differ by sex, race/ethnicity, and insurance type, with some differences attenuating in specific groups (e.g., Black infants). These findings underscore the importance of considering sociodemographic and biological context when evaluating infant growth outcomes. However, given the reduced sample sizes in subgroup analyses, these results should be interpreted cautiously, and larger, more diverse cohorts are needed to confirm and expand upon these observations.”
Comment 9
**Comment:** Beyond 1 year of age, infants’ diets are no longer primarily milk-based. The dietary patterns between ages 1–2 may have a greater impact on growth outcomes at 2 years than feeding types at 2 months. Therefore, the growth outcomes at 2 years may not fully support the study conclusions.
**Response:** We agree with the reviewer that after 12 months of age, dietary patterns diversify and milk/formula intake contributes less to overall energy consumption. Consequently, feeding type at 2 months may exert a diminishing influence on growth outcomes by 24 months. We have revised the Discussion to acknowledge this important limitation. At the same time, we believe it is still valuable to report 24-month outcomes, as early nutritional exposures may shape growth trajectories that persist into the second year of life. Our findings show that differences between feeding groups were attenuated but not eliminated at 24 months, suggesting that early formula type may have residual effects even after diet diversification.
**Manuscript Change:** Another limitation is that by 12 months of age, infants’ diets typically diversify, and dietary intake between ages 1 and 2 years may have a stronger impact on growth at 24 months than feeding type at 2 months. This dietary transition likely attenuated the associations we observed at 24 months. Nonetheless, reporting these outcomes remains important, as early nutritional exposures can influence growth trajectories beyond infancy, and we found that weight-for-age differences persisted into the second year of life.
Comment 10
**Comment:** Several studies have already analyzed the effects of certain formula types, such as hydrolyzed formulas or those with different protein contents, on infant growth. The novelty of this study does not seem to be clearly demonstrated.
**Response:** We thank the reviewer for raising this important point. Prior studies have examined specific formula types, particularly hydrolyzed or protein-modified formulas, often in small samples or clinical trial settings with limited follow-up duration. Our study adds novelty in several ways: (1) it uses a large, real-world cohort of over 5,500 infants derived from electronic health records, (2) it evaluates a spectrum of non-standard formulas reflective of actual clinical use, (3) it compares breastfeeding, standard formula, and non-standard formula simultaneously, and (4) it assesses growth at both 12 and 24 months, extending beyond the short-term adequacy outcomes required for formula approval. Together, these features provide new insights into how commonly used non-standard formulas may influence infant growth trajectories in a real-world U.S. population. We have expanded the Discussion to better emphasize these novel aspects.
**Manuscript Change:** Our study contributes novel insights to the literature on infant feeding and growth in several ways. First, unlike prior investigations that have focused on single formula subtypes or small trial populations, we evaluated a large, real-world cohort of over 5,500 infants using electronic health record data, which enhances generalizability to clinical practice. Second, we examined a broad spectrum of non-standard formulas, including lactose-reduced and partially hydrolyzed products that account for the majority of non-standard formula use in the United States, thereby reflecting the feeding patterns most relevant to families and clinicians. Third, our comparative design simultaneously assessed exclusive breastfeeding, standard formula, and non-standard formula, providing a comprehensive framework for understanding differences across feeding modalities. Finally, we assessed growth outcomes at both 12 and 24 months, extending beyond the short-term outcomes typically required for formula approval studies and offering a more longitudinal perspective on how early nutritional exposures may shape growth trajectories. Together, these features underscore the novelty of our study and its contribution to the evidence base guiding infant feeding decisions.
Comment 11
**Comment:** The manuscript is missing an ethical approval statement. The authors should, at a minimum, include the name of the ethics committee and the corresponding approval number in the main text.
**Response:** We thank the reviewer for pointing this out. We have added the Institutional Review Board approval details to the manuscript.
**Manuscript Change:** This study was approved by the University of Iowa Institutional Review Board (IRB No. 202506109).
Response to Reviewer 2
Introductory Summary
**Comment:** This is a retrospective study comparing infant anthropometric changes based on the type of milk intake determined at 2 months of age. Many clinical variables that might influence the infants’ growth were included in the GLMM method. Z-scores of anthropometric measurements served as dependent variables. Significant findings include higher weight, BMI, W/L, and length Z-scores for non-standard formula-fed infants compared to standard formula-fed infants at 12 months, as well as for the formula-fed group at 12 and 24 months compared to breastfed infants. These findings offer valuable information to physicians, aiding them in guiding parents to make informed decisions about their infant's nutrition.
**Response:** We thank the reviewer for their thoughtful summary of our study and for recognizing the potential clinical value of these findings.
Comment 1
**Comment:** I find it difficult to understand the last four variables in Table 1. The percentages of different milk types do not sum to 100%. If an infant is fed with formula, do you count it as both Cow’s milk and formula? I am curious why the percentages add up to more than 100%.
**Response**: We thank the reviewer for raising this important point. The percentages in Table 1 for the last four variables (cow’s milk, breast milk, formula, soy milk at 1 year of age) do not add up to 100% because families could report multiple milk types at the 1-year visit. For example, some infants were reported to be receiving both breast milk and cow’s milk, or both formula and cow’s milk. Thus, these categories are not mutually exclusive, and the percentages represent the proportion of infants in each feeding group who received each milk type, rather than summing to 100%. We have revised the Table 1 legend to clarify this.
**Manuscript Change** (Table 1 legend):
“Percentages for milk types at 1 year (breast milk, cow’s milk, formula, soy milk) do not add up to 100% because families could report more than one milk type at the 1-year visit (e.g., both breast milk and cow’s milk).”
Comment 2
**Comment:** Table 2 provides comprehensive information. The impact of non-standard versus standard formula varies across subgroup analyses. The difference disappears among Black infants. The effects on measurements differ by sex (at 12 months versus 24 months). While these findings are interesting, given the numerous subgroup analyses from a single dataset, shouldn’t you correct for false discovery rate (FDR)?
**Response:** We thank the reviewer for this important Comment. In Response, we have applied a Bonferroni correction to account for multiple comparisons across subgroup analyses. While this conservative approach reduced the number of statistically significant findings, the overall patterns of association remained consistent. We have added a statement in the Methods describing the application of the Bonferroni correction and have noted this adjustment in the Results and Discussion.
**Manuscript Change** (Methods, Statistical Analysis):
“Given the number of subgroup analyses performed, we applied a Bonferroni correction to adjust for multiple comparisons. Statistical significance was therefore interpreted conservatively in subgroup analyses.”
Comment 3
**Comment:** My primary concern is the effect that different milks have on anthropometrics, which are influenced by sex, ethnic group, age (12 months vs. 24 months), and insurance.
**Response:** We thank the reviewer for highlighting this important overarching concern. We agree that sex, race/ethnicity, age, and socioeconomic context are important determinants of growth outcomes and can influence how infant feeding type relates to anthropometrics. To address this, we adjusted for key covariates in all models (infant sex, maternal and infant health factors, and insurance type as a proxy for socioeconomic status). In addition, we performed subgroup analyses stratified by sex, race/ethnicity, insurance type, and birth weight, which are presented in Table 3. While some subgroup differences were attenuated (e.g., among Black infants), the overall pattern of higher growth z-scores in formula-fed infants compared to breastfed infants remained consistent. We have clarified this in the Discussion and emphasized that larger, more diverse cohorts are needed to further explore effect modification by sociodemographic and biological factors.
**Manuscript Change:** “Subgroup analyses suggested that the effects of non-standard versus standard formulas may differ by sex, race/ethnicity, and insurance type, with some differences attenuating in specific groups (e.g., Black infants). These findings underscore the importance of considering sociodemographic and biological context when evaluating infant growth outcomes. However, given the reduced sample sizes in subgroup analyses, these results should be interpreted cautiously, and larger, more diverse cohorts are needed to confirm and expand upon these observations.”
Reviewer 2 Report
Comments and Suggestions for Authors
This is a retrospective study comparing infant anthropometric changes based on the type of milk intake determined at 2 months of age. Many clinical variables that might influence the infants’ growth were included in the GLMM method. Z-scores of anthropometric measurements served as dependent variables. Significant findings include higher weight, BMI, W/L, and length Z-scores for non-standard formula-fed infants compared to standard formula-fed infants at 12 months, as well as for the formula-fed group at 12 and 24 months compared to breastfed infants. These findings offer valuable information to physicians, aiding them in guiding parents to make informed decisions about their infant's nutrition.
I find it difficult to understand the last four variables in Table 1. The percentages of different milk types do not sum to 100%. If an infant is fed with formula, do you count it as both Cow’s milk and formula? I am curious why the percentages add up to more than 100%.
Table 2 provides comprehensive information. The impact of non-standard versus standard formula varies across subgroup analyses. The difference disappears among black infants. The effects on measurements differ by sex (at 12 months versus 24 months). While these findings are interesting, given the numerous subgroup analyses from a single dataset, shouldn’t you correct for false discovery rate (FDR)?
My primary concern is the effect that different milks have on anthropometrics, which are influenced by sex, ethnic group, age (12 months vs. 24 months), and insurance.
Author Response
Response to Reviewer 2
Introductory Summary
**Comment:** This is a retrospective study comparing infant anthropometric changes based on the type of milk intake determined at 2 months of age. Many clinical variables that might influence the infants’ growth were included in the GLMM method. Z-scores of anthropometric measurements served as dependent variables. Significant findings include higher weight, BMI, W/L, and length Z-scores for non-standard formula-fed infants compared to standard formula-fed infants at 12 months, as well as for the formula-fed group at 12 and 24 months compared to breastfed infants. These findings offer valuable information to physicians, aiding them in guiding parents to make informed decisions about their infant's nutrition.
**Response:** We thank the reviewer for their thoughtful summary of our study and for recognizing the potential clinical value of these findings.
Comment 1
**Comment:** I find it difficult to understand the last four variables in Table 1. The percentages of different milk types do not sum to 100%. If an infant is fed with formula, do you count it as both Cow’s milk and formula? I am curious why the percentages add up to more than 100%.
**Response**: We thank the reviewer for raising this important point. The percentages in Table 1 for the last four variables (cow’s milk, breast milk, formula, soy milk at 1 year of age) do not add up to 100% because families could report multiple milk types at the 1-year visit. For example, some infants were reported to be receiving both breast milk and cow’s milk, or both formula and cow’s milk. Thus, these categories are not mutually exclusive, and the percentages represent the proportion of infants in each feeding group who received each milk type, rather than summing to 100%. We have revised the Table 1 legend to clarify this.
**Manuscript Change** (Table 1 legend):
“Percentages for milk types at 1 year (breast milk, cow’s milk, formula, soy milk) do not add up to 100% because families could report more than one milk type at the 1-year visit (e.g., both breast milk and cow’s milk).”
Comment 2
**Comment:** Table 2 provides comprehensive information. The impact of non-standard versus standard formula varies across subgroup analyses. The difference disappears among Black infants. The effects on measurements differ by sex (at 12 months versus 24 months). While these findings are interesting, given the numerous subgroup analyses from a single dataset, shouldn’t you correct for false discovery rate (FDR)?
**Response:** We thank the reviewer for this important comment. In response, we have applied a Bonferroni correction to account for multiple comparisons across subgroup analyses. While this conservative approach reduced the number of statistically significant findings, the overall patterns of association remained consistent. We have added a statement in the Methods describing the application of the Bonferroni correction and have noted this adjustment in the Results and Discussion.
**Manuscript Change** (Methods, Statistical Analysis):
“Given the number of subgroup analyses performed, we applied a Bonferroni correction to adjust for multiple comparisons. Statistical significance was therefore interpreted conservatively in subgroup analyses.”
Comment 3
**Comment:** My primary concern is the effect that different milks have on anthropometrics, which are influenced by sex, ethnic group, age (12 months vs. 24 months), and insurance.
**Response:** We thank the reviewer for highlighting this important overarching concern. We agree that sex, race/ethnicity, age, and socioeconomic context are important determinants of growth outcomes and can influence how infant feeding type relates to anthropometrics. To address this, we adjusted for key covariates in all models (infant sex, maternal and infant health factors, and insurance type as a proxy for socioeconomic status). In addition, we performed subgroup analyses stratified by sex, race/ethnicity, insurance type, and birth weight, which are presented in Table 3. While some subgroup differences were attenuated (e.g., among Black infants), the overall pattern of higher growth z-scores in formula-fed infants compared to breastfed infants remained consistent. We have clarified this in the Discussion and emphasized that larger, more diverse cohorts are needed to further explore effect modification by sociodemographic and biological factors.
**Manuscript Change:** “Subgroup analyses suggested that the effects of non-standard versus standard formulas may differ by sex, race/ethnicity, and insurance type, with some differences attenuating in specific groups (e.g., Black infants). These findings underscore the importance of considering sociodemographic and biological context when evaluating infant growth outcomes. However, given the reduced sample sizes in subgroup analyses, these results should be interpreted cautiously, and larger, more diverse cohorts are needed to confirm and expand upon these observations.”
Round 2
Reviewer 1 Report
Comments and Suggestions for Authors
The authors’ revisions are acceptable. The study is novelty and clinical significance, and I recommend it for publication.
Reviewer 2 Report
Comments and Suggestions for Authors
I am more than happy with the response.